# Anticoagulant Treatment in Patients with AF and Very High Thromboembolic Risk in the Era before and after the Introduction of NOAC: Observation at a Polish Reference Centre

**DOI:** 10.3390/ijerph20126145

**Published:** 2023-06-16

**Authors:** Bernadetta Bielecka, Iwona Gorczyca-Głowacka, Agnieszka Ciba-Stemplewska, Beata Wożakowska-Kapłon

**Affiliations:** 11st Clinic of Cardiology and Electrotherapy, Świętokrzyskie Cardiology Centre, 25-736 Kielce, Poland; bielecka.bernadetta@gmail.com (B.B.); bw.kaplon@poczta.onet.pl (B.W.-K.); 2Collegium Medicum, Jan Kochanowski University, 25-369 Kielce, Poland; aciba@interia.pl; 3Clinic of Internal Medicine, 25-736 Kielce, Poland

**Keywords:** atrial fibrillation, non-vitamin K antagonist oral anticoagulants, oral anticoagulants

## Abstract

Atrial fibrillation (AF) is associated with an increased risk of stroke. Therefore, patients with AF require appropriate management and anticoagulant therapy. To balance therapy risks and benefits, oral anticoagulants (OAC) treatment should be ‘tailored’ in patients at a high risk of stroke and bleeding. However, some studies have demonstrated that certain groups of patients do not receive anticoagulants despite the high risk of stroke or thromboembolism. The study aimed to analyse therapeutic methods of stroke prevention in very high-risk patients (CHA_2_DS_2_-VASc score of ≥5 in men and ≥6 in women), identify factors predisposing against the use of OACs and assess the administration of anticoagulants before the introduction of non-vitamin K antagonist OAC (NOAC) in 2004–2011 and beyond (years 2012–2019). The analysis covered 2441 patients with AF at a very high thromboembolic risk who were hospitalised in a reference cardiological centre from 2004 to 2019. Data concerning patients’ sex, age, comorbidities, type of AF, renal and echocardiographic parameters, reasons for hospitalisation and applied treatment were collected from medical records. HAS-BLED, CHADS_2_, and CHA_2_DS_2_-VASc scores were calculated for all patients. The treatment with oral anticoagulants was compared in the entire population over 2004–2011 and 2012–2019. In this study, a fifth of patients were not treated with OAC. Most patients hospitalised in the years 2012–2019 were treated with OAC. The predictors of not using OAC turned out to be: age of >74 years, heart failure, cancer, paroxysmal AF, and acute coronary syndrome (ACS) or elective coronary angiography/percutaneous coronary intervention (PCI) as a reason for hospitalisation. The introduction of NOAC was associated with a decline in the use of VKA (from 62% to 19.1%) and APT (from 29.1% to 1.3%). This study outlines reasons to initiate OAC treatment in very high-risk patients in clinical practice.

## 1. Introduction

Atrial fibrillation (AF) drastically increases the risk of stroke. Therefore, patients with this disorder require considerate management and appropriate anticoagulant therapy [1]. Up to 40% of patients with acute ischemic stroke are diagnosed for the first time with atrial fibrillation [2]. The use of oral anticoagulants (OAC) [including vitamin K antagonists (VKAs) or non-vitamin K antagonist OACs (NOACs)] diminishes the risk of AF-related thromboembolic strokes and patients’ all-cause mortality [3]. Guidelines concerning the management of AF recommend the administration of OAC in all patients with AF burdened with stroke risk factors [3]. However, such an approach should be appropriately ‘tailored’ in patients at a high risk of stroke and bleeding to balance therapy risks and benefits. The incidence of ‘unmodified’ risk factors in patients with AF makes the administration of OAC challenging in this population [4]. The optimal use of OAC in clinical practice also requires correcting modifiable bleeding risk factors, including inappropriate VKA treatment reflected by labile INR (International Normalized Ratio), poorly controlled hypertension, or the use of, e.g., non-steroidal anti-inflammatory drugs [5,6].

Risk score scales are broadly used in clinical settings to support deciding on appropriate treatment [7]. The CHA_2_DS_2_-VASc score is recommended to assist in recognition of individuals at increased thromboembolic risk, while the use of HAS-BLED is advocated to evaluate bleeding risk [8,9]. These scores should be used to identify patients with a high risk of bleeding and to correct potentially modifiable bleeding risk factors, but not to withhold OAC treatment. According to guidelines, anticoagulation therapy is a Class I recommendation for patients whose CHA_2_DS_2_-VASc score is equal to or higher than two (males) or three (females). However, risk factors in the model do not carry equal weight; thus, this score may lack precision, especially in high-risk patients [10,11].

Many studies have demonstrated that certain groups of patients do not receive anticoagulants (even those without contraindications) despite the high risk of stroke or thromboembolism [1,12,13,14]. Some patients are treated with antiplatelet agents instead, which does not reduce haemorrhagic risk, while others may be prescribed anticoagulant therapy but fail to take it. Distrust in risk stratification, making treatment decisions on patients’ individual risk factors, and the fear of bleeding may underlie some physicians’ reluctance to prescribe anticoagulants [1,7,15].

Therefore, we aimed to analyse therapeutic methods of stroke prevention in very high-risk patients (CHA_2_DS_2_-VASc score of ≥5 in men and ≥6 in women), identify factors predisposing against the use of OACs in this group of patients and assess the administration of anticoagulants before the introduction of NOAC in 2004–2011 (new oral anticoagulants) and after their introduction (years 2012–2019).

## 2. Materials and Methods

### 2.1. Study Group

This retrospective study included patients with AF hospitalised from January 2004 to December 2019 at Świętokrzyskie Cardiology Centre, which is the largest reference centre in Świętokrzyskie Voivodeship. The analysis covered 2441 patients at a very high thromboembolic risk—1270 women with a CHA_2_DS_2_-VASc score of ≥6 and 1171 men with a score of ≥5 (Figure 1). The enrolled patients were >18 years and were either emergency or elective hospital patients. Patients with incomplete data on treatment, with valvular disease, those who died during hospitalisation, women with CHA_2_DS_2_-VASc score of <6 points, and men with CHA_2_DS_2_-VASc score < 5 points were excluded from this study. Above these cut-off points, a significant increase in thromboembolic risk is observed [16].

### 2.2. Methods

Data concerning patients’ sex, age, comorbidities, type of AF, renal and echocardiographic parameters, reasons for hospitalisation, and applied treatment were collected from both pared and electronic discharge cards of patients hospitalised in the Świętokrzyskie Center of Cardiology. Anaemia was defined as a decrease in haemoglobin < 12 g/dL, and thrombocytopenia was defined as a decrease in platelets (PLT) < 150,000/μL.

The risk of bleeding was defined using the HAS-BLED score, which includes arterial hypertension, impaired renal/liver function, stroke, bleeding, labile INR, older age (>65 years), drugs, and alcohol [5].

Glomerular filtration rate (GFR), used to assess patients’ renal function, was calculated using the CKD-EPI equation (Chronic Kidney Disease Epidemiology Collaboration).

AF was diagnosed based on the European Society of Cardiology’s definition, according to which arrhythmia can be identified using an electrocardiogram that shows irregular atrial rhythm that lasts longer than 30 s [17].

Paroxysmal AF was defined in agreement with the American Heart Association and American College of Cardiology guidelines [18] as an episode of intermittent, irregular heart rhythm lacking P waves in electrocardiography that lasts over 30 s and terminates spontaneously or within seven days of treatment. Persistent AF was diagnosed in patients whose abnormal heart rhythms lasted over a week [18]. Permanent AF was diagnosed according to 2016 European Society of Cardiology guidelines [19] in patients in whom rhythm control interventions were not pursued. Non-permanent AF is a non-permanent atrial fibrillation, i.e., paroxysmal and persistent. In our work, it is the sum of paroxysmal and persistent AF.

The study was approved by the Ethics Committee of the Świętokrzyska Medical Chamber in Kielce (Approval No. 12/2011; 2/2023). The Committee waived the requirement of obtaining informed consent from the patients.

### 2.3. Assessment of the Thromboembolic Risk

CHADS_2_ and CHA_2_DS_2_-VASc scores were calculated for all patients. The use of both scores: the CHADS_2_ score and its modification containing new risk factors—CHA_2_DS_2_-VASc (developed in 2010) is associated with the study also included patients hospitalised from 2004 to 2019. After 2010, the use of the CHA_2_DS_2_-VASc scale was recommended. The CHADS_2_ scale included heart failure (HF), hypertension, age of ≥75 years, diabetes mellitus, and a history of thromboembolic events. In the presented study, the CHA_2_DS_2_-VASc scale was used to consider additional risk factors recommended by the ESC guidelines [20]. The CHA_2_DS_2_-VASc scale included congestive heart failure, hypertension, age of ≥75 years, diabetes, stroke/transient ischemic attack (TIA)/thromboembolic event, vascular disease, age of 65–74 years, and female gender.

### 2.4. Management of Antithrombotic Therapy

The anticoagulant therapy prescribed at the time of discharge from the hospital was evaluated in this study. Three treatment regimens were defined: OAC ± antiplatelet drug (APT), APT alone, low molecular weight heparin (LMWH), and lack of anticoagulant therapy. The OAC therapy included VKAs, apixaban, dabigatran, and rivaroxaban alone or with APT. Edoxaban has been registered in Europe as a drug to prevent thromboembolic complications in patients with AF. However, it is not available in Poland. The APT group of drugs included acetylsalicylic acid (ASA) and/or clopidogrel, ticagrelor, and prasugrel.

The anticoagulation treatment of patients treated with OAC and those not receiving oral anticoagulants (non-OAC) was compared in the entire population in the years 2004–2011 (before the introduction of NOACs into clinical practice) and in 2012–2019 (after the introduction of new anticoagulants).

### 2.5. Statistical Analysis

Statistical analyses were performed using the IBM SPSS Statistics version 26 package to answer the research questions. Mann–Whitney U tests were used due to significant discrepancies in the number of patients between the compared groups, and also chi-square tests of independence were carried out. Fisher’s exact tests were performed if the assumptions of the chi-square tests regarding the number of expected observations were unmet. The classical threshold of α = 0.05 (*p* < 0.05) was adopted as the level of statistical significance. A series of one-way logistic regression analyses were performed to investigate the factors contributing to the non-use of OACs by patients. A multivariate regression analysis was performed to assess why OAC was not used. Significant variables obtained in univariate models were included as predictors.

## 3. Results

### 3.1. Characteristics of the Study Group

The study group included a similar number of females and males (52% females). The mean age of patients was 78.7 ± 7.2 years; most were over the age of 74 years (n = 1906; 78.1%).

The most frequent comorbidities included: arterial hypertension in 2259 (92.5%) patients, heart failure (n = 2098; 85.9%), and vascular disease (n = 1697; 69.5%). The eGFR of 77.1% (n = 1872) of patients was below 60 mL/min/1.73m^2^. Non-permanent AF was more common in the patients not using OAC (51.4% vs. 44.5%) (*p* = 0.009). Table 1 presents patients’ characteristics.

Patients in the non-OAC group were significantly older (79.6 ± 6.9 vs. 78.6 ± 7.2), were more likely to have such comorbidities as heart failure (90.1% vs. 85%, *p* = 0.005), cancer (8.3% vs. 4.2%, *p* < 0.001) and paroxysmal AF (49.3% vs. 38.3%, *p* < 0.001) compared to those treated with OAC. The mean CHA_2_DS_2_-VASc score in all patients was 6.1 ± 1 points. Patients using OAC had a higher bleeding risk than those not using (2.5 ± 0.7 vs. 2.4 ± 0.7). Moreover, OACs were more frequently used by patients admitted to the hospital due to electrical cardioversion, heart failure, AF without any procedures and other causes compared to patients admitted for planned coronary angiography, percutaneous coronary interventions (PCI), or acute coronary syndromes (ACS).

Baseline characteristics of various CHA_2_DS_2_-VASc scale groups of females are presented in Appendix A and males in Appendix A.

Figure 2 shows the CHA_2_DS_2_-VASc score-derived OAC prescription rate, and Figure 3 shows the prescription of OACs in patients grouped based on the HAS-BLED score.

In this study, we also compared the frequency of OAC prescriptions in 2004–2011 and 2012–2019, summarised in Table 2.

Over 2012–2019, OAC was more frequently prescribed to patients with vascular disease and peripheral artery disease (PAD) compared to the years 2004–2011 (76.2% vs. 52.6%, *p* < 0.001, and 17.8% vs. 3%, *p* < 0.001, respectively). Moreover, OAC was more commonly administered to patients with myocardial infarction, PCI, and coronary artery bypass grafting (CABG) from 2012–2019. In turn, more patients with previous stroke (38.5% vs. 28.6%, *p* < 0.001) and peripheral thromboembolic events (7.2% vs. 3.8%, *p* = 0.002) were treated with OAC in 2004–2011.

The subjects (both those using and not using OAC) hospitalised in the years 2004–2011 were characterised by a lower risk of bleeding than the subjects hospitalised in the years 2012–2019 (HAS-BLED 2.4 ± 0.6 vs. 2.5 ± 0.7, *p* < 0.001 for OAC and 2.2 ± 0.5 vs. 2.6 ± 0.9, *p* < 0.001 for non-OAC). Only a small percentage of patients with a history of bleeding were treated with OAC in both analysed periods. However, after the introduction of NOAC, more such patients received treatment (1.7% vs. 4.2%) (*p* = 0.01).

The analysis of the total population of patients revealed that out of 2441 patients, 2005 (82.1%) were prescribed OAC, 294 (12%) patients received APT, 60 (2.5%) patients received LMWH, and 82 (3.4%) patients remained without treatment (Appendix A).

In the years 2004–2011, 470 (62.6%) patients received OAC, 240 (32%) patients received APT, 19 (2.5%) patients received LMWH, and 22 (2.9%) patients remained untreated. In the years 2012–2019, 1535 (90.8%) patients received OAC, 54 (3.2%) patients received APT, 41 (2.4%) patients—LMWH, while 60 (3.6%) patients remained without treatment.

Following the introduction of NOAC, the treatment mode changed. In 2012–2019, among patients treated with OAC, 1101 (54.9%) received VKA and 904 (45.1%) NOAC. 207 (22.9%) of patients treated with NOACs received apixaban, 410 (45.4%) used dabigatran, and 287 (31.7%) patients received rivaroxaban.

Among patients at a very high risk of thromboembolism in the years 2004–2019, changes in treatment regimens were observed—the administration of VKA decreased from 62% to 19.1%, APT from 29.1% to 1.3%, LMWH from 2.5% to 0.6%. From 2012 to 2019, the use of NOAC in patients with a very high thromboembolic risk increased from 4% to 79%. Figure 4 presents the proportion of patients treated with OAC/non-OACs during the observed years.

### 3.2. Predictors of Non-OAC Treatment in Very High-Risk Patients

Univariable logistic regression analysis revealed numerous predictors associated with not prescribing OAC (Appendix A).

Table 3 demonstrates predictors of OAC non-use in the multivariable model, including the age of >74 years (OR, 1.78; 95% CI, 1.32–2.42; *p* < 0.001), heart failure (OR, 2.05; 95% CI, 1.41–2.98; *p* < 0.001), cancer (OR, 2.16; 95% CI, 1.38–3.40; *p* = 0.001), paroxysmal AF (OR, 1.69; 95% CI, 1.34–2.14; *p* < 0.001) and ACS or elective coronary angiography/PCI as a reason for hospitalisation (OR, 2.41; 95% CI, 1.81–3.22; *p* < 0.001). Hospitalisation in 2012–2019 compared to the hospitalisation in 2004–2011 was a negative predictor of not using OAC (OR, 0.17; 95% CI, 0.14–0.22; *p* < 0.001). This analysis showed that the probability of OAC non-use was significantly higher in patients over 74 years compared to younger individuals. Patients with HF, tumours (two times lower probability for both), paroxysmal AF, and patients hospitalised due to ACS or elective coronary angiography/PCI were less likely to receive OAC treatment (OR = 2.41).

The Hosmer and Lemeshow test results showed that the model fits the data well: χ2(8) = 9.26; *p* = 0.321. The model explains 23% of the variance of not using OAC (Nagelkerke R2 = 0.23). There were 83.4% correct matches.

## 4. Discussion

This study provides insight into treating patients with very high thromboembolic risk with anticoagulants. In this study, about a fifth of patients were not treated with OAC. Most patients treated with OAC were hospitalised during 2012–2019 (after the introduction of NOAC). Finally, we identified factors that were significantly associated with not prescribing OAC.

AF is a main stroke risk factor; therefore, patients with this disorder require considerate management and appropriate anticoagulant therapy [1]. In patients with a very high risk of stroke and bleeding, the therapy should be tailored based on balancing treatment risks and benefits to obtain favourable outcomes [3]. We observed that nearly one-fifth of patients were not treated with OAC. As much as 44.3% of patients with a CHA_2_DS_2_-VASc score of 6 and 2.8% with a CHA_2_DS_2_-VASc score of 9 did not receive OAC. Approximately 31.7% of individuals in the untreated group had a history of stroke. Similarly, in the study of Lee et al. [1], a group of patients with AF was not treated with anticoagulants though their CHA_2_DS_2_-VASc score indicated a high risk of a stroke or thromboembolism. We did not observe differences in a thromboembolic risk assessed based on CHADS_2_ and CHA_2_DS_2_-VASc between OAC-treated and untreated groups. Similarly, Szpotowicz et al. [21] found that the CHA_2_DS_2_-VASc score did not predict OAC use in the studied cohort of patients with AF.

Moreover, we observed that 3.4% of patients with AF in our cohort did not receive OAC, low-molecular-weight heparin (LMWH), or antiplatelet therapy. Similarly, other studies reported that some high-risk AF patients without contraindications did not receive thromboprophylaxis [13,22]. It is frequently difficult to decide whether the lack of anticoagulant treatment was associated with the physician’s decision or the reluctance of the patient [1]. Extensive retrospective studies and meta-analyses of patients with AF showed that despite being considered high-risk of stroke or thromboembolism, a large group did not receive anticoagulants, even those without contraindications [1,12,13]. The United States Medicare database analysis revealed that 51.3% of patients diagnosed with AF were not prescribed an OAC during the mean follow-up of 2.4 years [23]. In the Spanish study, which focused on patients with non-valvular AF and moderate-to-high risk of stroke, 20% of analysed individuals did not receive OAC even if there were no clear contraindications to OACs [14]. It could not be ruled out that some patients were prescribed but failed to redeem it. It also appears that some physicians are reluctant to prescribe anticoagulants due to a lack of trust in the risk stratification and the fear of bleeding [1]. Based on Optum’s De-identified Integrated Claims-ERH data, Guo et al. showed that 19% of patients with AF prescribed did not buy the drug [24].

The incidence of AF increases with age; however, the results of studies suggest that older patients (especially those over the age of ≥90 years) with a perceived high risk for bleeding (e.g., chronic kidney disease) are often refused OACs [15]. A large retrospective study of a cohort of patients with nonvalvular AF revealed that physicians make their treatment decisions on the constellation of patient’s risk factors, such as bleeding risk factors (including drugs and hypertension within the year before diagnosis), age and history of major bleeding, not on values of the risk scores, even though international guidelines do not point to bleeding risk factors as a contraindication to treatment [7]. In turn, Volgman et al. [25] found that syncope and fall-related injuries are the most frequent causes of non-prescription of OAC among elderly patients (75 years or above).

Making a decision based on individual risk assessment in this group of patients could be associated with the fact that the CHA_2_DS_2_-VASc score identifies high-risk patients only with modest accuracy [1]. Apart from some ‘unmodified’ factors, also modifiable bleeding risk factors, including poorly controlled hypertension, alcohol abuse, inaccurate VKA, and co-treatment with antiplatelet or non-steroidal anti-inflammatory drugs, must be taken into consideration while prescribing OAC therapy [3,5,6]. In our study, over 40% of OAC-treated patients had high bleeding risk (HAS-BLED of ≥3), and only a small percentage of patients had a history of bleeding and/or HAS-BLED of ≥5 confirms that this factor is of importance for physicians while deciding on OAC treatment. National and international guidelines suggest that patients with a risk of bleeding should be more carefully monitored and undergo more regular check-ups, and a strategy to modify risk factors ought to be introduced (if possible) instead of depriving them of OAC treatment [26,27]. A systematic review and meta-analysis of stroke risk score performance, which included 6,267,728 patients with AF, revealed that newer risk scores and updates compared with the CHADS_2_ and CHA_2_DS_2_-VASc showed improved discrimination [28]. Also, ESC guidelines (2020) [29] suggest that more complex clinical scales also involving other risk factors, such as Global Anticoagulant Registry in the FIELD—Atrial Fibrillation (GARFIELD-AF), Anticoagulation and Risk Factors in Atrial Fibrillation (ATRIA), Intermountain Risk Score, or ABC-stroke (Age, Biomarkers, Clinical history) improved the assessment of stroke risk to a small but statistically significant extent and might therefore be considered for use in clinical practice.

Our study also demonstrated the increase in the OAC prescription rate after the introduction of NOAC compared to the earlier analysed period (2004–2011). Similarly, other nationwide cohort studies have reported increased use of NOACs for stroke prevention in AF following the introduction of NOACs into clinical practice [30,31].

In our study, we also identified predictors of OAC non-use. Our multivariable model revealed that age of >74 years, HF, cancer, paroxysmal AF, and ACS or elective coronary angiography/PCI as a reason for hospitalisation decreased the odds of the treatment with OAC. Moreover, in our study, 55.5% of patients with permanent AF received OAC compared to 38.3% of patients with paroxysmal AF. This is in agreement with the results of other studies, which have found that the type of AF may play a role in deciding on the treatment. Anticoagulant therapy was suggested to be prescribed more frequently in patients with permanent AF compared with patients with paroxysmal AF, even though the guidelines advocate anticoagulation irrespective of AF type [32,33]. In turn, the results of the study performed by Besford et al. [7] demonstrated that bleeding risk factors, including hypertension within one year before diagnosis, age at diagnosis as well as the history of major bleeding, and use of drugs that increase the risk of bleeding in the year before diagnosis were the most important factors in the treatment decision. Similarly, the Health Improvement Network database pointed to very old age. However, also younger age, female sex, a HAS-BLED score of ≥3, a history of intracranial bleeding, higher Charlson Comorbidity Index, falls, and polypharmacy as factors that were significantly associated with a lower likelihood of receiving OAC [34]. The United States-based ambulatory cardiology registry analysis shows that apart from age, stroke, and bleeding risk, reversible AF aetiology, renal, liver, or vascular disease are factors associated with OAC non-prescription [35]. However, according to Lubitz et al. [35], the use of antiplatelet drugs translated into the highest risk of OAC non-prescription.

In 2012–2019, the use of NOAC in patients at a very high thromboembolic risk increased from 4% to 79%, and dabigatran was the most frequently prescribed drug, followed by rivaroxaban and apixaban. We also observed that the introduction of NOAC translated into significantly decreased use of VKA (from 62% to 19.1%) and APT (to 3.2% from 32%) in 2012–2019 compared to 2004–2011. Vitamin K antagonists and NOACs are recommended in patients with AF without moderate or severe mitral valve stenosis or prosthetic mechanical heart valves [3]. Currently, most guidelines advocate using NOACs as the first line OAC; however, caution is required in some groups of patients [3]. NOACs offer a greater net clinical benefit than VKAs; in most cases, they are now first-line drugs [3,31].

## 5. Conclusions

The results from this study can improve our understanding concerning the decision to initiate OAC treatment in very high-risk patients in clinical practice. We demonstrated that nearly 20% of patients with AF in our study did not receive appropriate treatment. The increase in the use of OAC was observed in patients hospitalised in the years 2012–2019, which could be associated with the introduction of NOAC. We also observed that the introduction of NOAC translated into a lower rate of administration of VKA and APT. Finally, we identified that age of >74 years, HF, cancer, paroxysmal AF, and ACS or elective coronary angiography/PCI as reasons for hospitalisation were significantly associated with not prescribing OAC.

## 6. Limitations

The limitation of this study present study includes the retrospective nature of data collected in one centre. However, since it is a reference clinic in the voivodeship, it admits ambulatory patients from other hospitals from Świętokrzyskie Voivodeship and surrounding voivodeships. The retrospective nature of this study and no access to data from cardiology outpatients’ clinics are also the reasons why we lack the assessment of patients several months after hospitalisation. Thus, we could not assess whether stroke patients discharged without OACs were administered such treatment after evaluation (e.g., after one month). However, most patients had a stroke in the past; thus, stroke was not the reason for hospitalisation in our study. This study also lacks information on the exact reason concerning the non-prescribing of OAC. Moreover, the indications for OAC use differed slightly in particular years (2004–2019), and the exact starting point of NOAC introduction in Poland is difficult to assess. All patients were treated following the most recently published guidelines concerning AF in force at the time of their admission to our centre. Our study lacks data on edoxaban since this drug is currently unavailable in Poland.

Despite the aforementioned limitations, we believe that the presented data provide useful and reliable insight into real clinical practice and may increase the awareness of the need to develop strategies to reduce the underuse of OAC by patients with AF. This is a unique study of patients with a very high thromboembolic risk, which may increase the awareness of those clinicians who are afraid of using anticoagulant treatment in this group of patients. Only a few similar studies assess the non-use of OAC in such a burdened group of patients.

## Figures and Tables

**Figure 1 ijerph-20-06145-f001:**
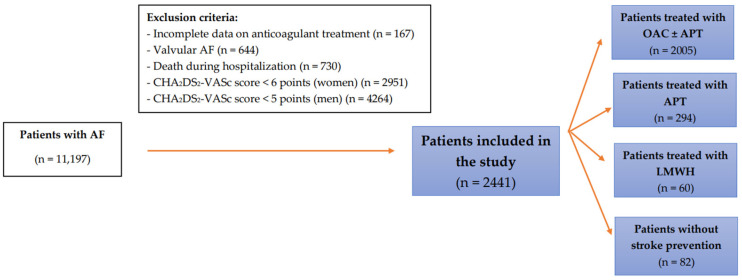
The flow chart of the study. Abbreviations: AF, atrial fibrillation; APT, antiplatelet drugs; OAC, oral anticoagulant therapy; LMWH, low molecular weight heparin.

**Figure 2 ijerph-20-06145-f002:**
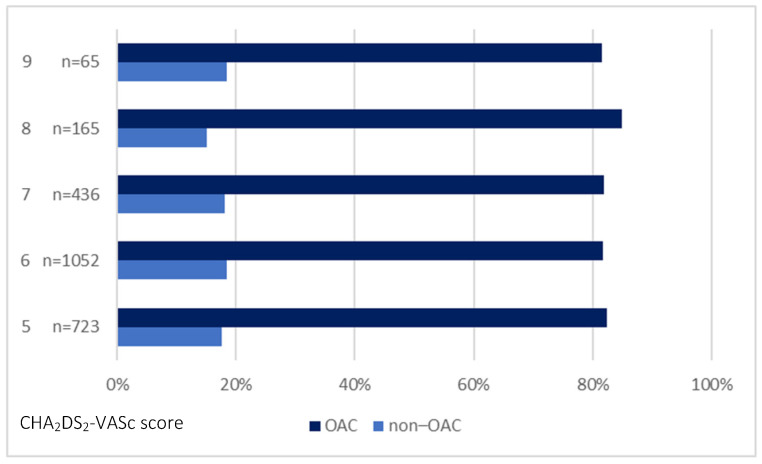
The prescription of OAC based on the CHA2DS2-VASc score in the entire population. Abbreviation: OAC, oral anticoagulant.

**Figure 3 ijerph-20-06145-f003:**
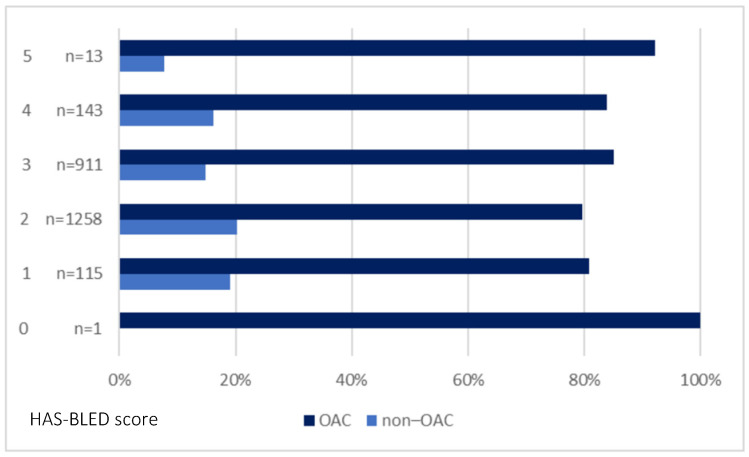
The prescription of OAC based on the HAS-BLED score in the entire population. Abbreviation: OAC, oral anticoagulant.

**Figure 4 ijerph-20-06145-f004:**
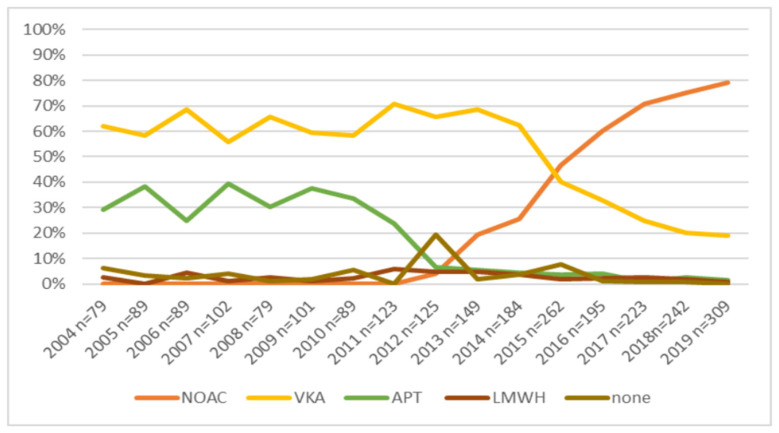
Stroke prophylaxis in patients with a very high risk of thromboembolic events with AF. Abbreviation: APT, antiplatelet drug; LMWH, low mass weight heparin; NOAC, non-vitamin K antagonist oral anticoagulants; VKA, vitamin K antagonists.

**Table 1 ijerph-20-06145-t001:** Baseline characteristics of the study groups (OAC vs. non-OAC).

	All Patients n = 2441	OAC n = 2005	Non-OAC n = 436	*p*
Sex (female), n (%)	1270 (52)	1043 (52)	227 (52.1)	0.987
Age, mean (SD)	78.7 (7.2)	78.6 (7.2)	79.6 (6.9)	0.007
Age < 65, n (%)	71 (2.9)	58 (2.9)	13 (3)	0.920
Age 65–74, n (%)	(19)	408 (20.3)	56 (12.8)	<0.001
Age > 74, n (%)	1906 (78.1)	1539 (76.8)	367 (84.2)	0.001
Heart failure, n (%)	2098 (85.9) 464	1705 (85)	393 (90.1)	0.005
Arterial hypertension, n (%)	2259 (92.5)	1861 (92.8)	398 (91.3)	0.269
Vascular disease, n (%)	1697 (69.5)	1417 (70.7)	280 (64.2)	0.008
Diabetes mellitus, n (%)	1288 (52.8)	1057 (52.7)	231 (53)	0.920
Previous stroke, n (%)	773 (31.7)	635 (31.7)	138 (31.7)	0.994
Previous TIA, n (%)	144 (5.9)	126 (6.3)	18 (4.1)	0.083
Peripheral thromboembolic events, n (%)	105 (4.3)	93 (4.6)	12 (2.8)	0.079
Myocardial infarction, n (%)	853 (35.2)	718 (35.8)	135 (31)	0.054
Stable CAD, n (%)	841 (34.5)	714 (35.6)	127 (29.1)	0.010
PCI, n (%)	530 (21.7)	464 (23.1)	66 (15.1)	<0.001
CABG, n (%)	238 (9.8)	216 (10.8)	22 (5)	<0.001
PAD, n (%)	321 (13.2)	287 (14.3)	34 (7.8)	<0.001
Bleeding, n (%)	96 (3.9)	73 (3.6)	23 (5.3)	0.112
Peptic ulcer disease, n (%)	83 (3.4)	62 (3.1)	21 (4.8)	0.072
Cancer, n (%)	121 (5)	85 (4.2)	36 (8.3)	<0.001
Thrombocythemia, n (%)	435 (17.8)	347 (17.3)	88 (20.2)	0.155
Anaemia, n (%)	647 (26.5)	521 (26)	126 (28.9)	0.212
Dialysis, n (%)	6 (0.2)	4 (0.2)	2 (0.5)	0.292 *
Type of AF, n (%)				
Paroxysmal	982 (40.2)	767 (38.2)	215 (49.3)	<0.001
Persistent	135 (5.6)	126 (6.3)	9 (2.1)	<0.001
Permanent	1324 (54.2)	1112 (55.5)	212 (48.6)	0.009
Non-permanent (paroxysmal + persistent)	1117 (45.8)	893 (44.5)	224 (51.4)	0.009
Thromboembolic risk				
CHADS_2_, mean (SD)	3.9 (0.9)	3.9 (0.9)	3.9 (0.9)	0.361
CHA_2_DS_2_-VASc, mean (SD)	6.1 (1)	6.1 (1)	6.1 (1)	0.906
CHA_2_DS_2_-VASc = 5, n (%)	723 (29.6)	596 (29.7)	127 (29.1)	0.804
CHA_2_DS_2_-VASc = 6, n (%)	1052 (43.1)	859 (42.8)	193 (44.3)	0.587
CHA_2_DS_2_-VASc = 7, n (%)	436 (17.9)	357 (17.8)	79 (18.1)	0.877
CHA_2_DS_2_-VASc = 8, n (%)	165 (6.8)	140 (7)	25 (5.7)	0.347
CHA_2_DS_2_-VASc = 9, n (%)	65 (2.7)	53 (2.6)	12 (2.8)	0.898
Bleeding risk				
HAS-BLED, mean (SD)	2.5 (0.7)	2.5 (0.7)	2.4 (0.7)	0.002
HAS-BLED ≥ 3, n (%)	1067 (43.7)	908 (45.3)	159 (36.5)	0.001
HAS-BLED ≥ 5, n (%)	13 (0.5)	12 (0.6)	1 (0.2)	0.485 *
Laboratory test results				
eGFR (mL/min/1.73m^2^), mean (SD)	n = 2429 49.1 (15.9)	n = 1997 49.3 (15.6)	n = 432 48.7 (17.2)	0.434
eGFR ≥ 60 mL/min/1.73 m^2^, n (%)	557 (22.9)	459 (23)	98 (22.7)	0.893
eGFR 59–45 mL/min/1.73 m^2^, n (%)	875 (36)	719 (36)	156 (36.1)	0.966
eGFR 44–30 mL/min/1.73 m^2^, n (%)	736 (30.3)	614 (30.7)	122 (28.2)	0.304
eGFR 29–15 mL/min/1.73 m^2^, n (%)	237 (9.8)	188 (9.4)	49 (11.4)	0.221
eGFR < 15 mL/min/1.73 m^2^, n (%)	24 (1)	17 (0.9)	7 (1.6)	0.174 *
Echocardiography				
EF (%), mean (SD)	n = 1895 45.8 (12.6)	n = 1567 45.9 (12.9)	n = 328 45.2 (11.5)	0.132
EF ≥ 50%, n (%)	920 (48.5)	776 (49.5)	144 (43.9)	0.064
EF 49–41%, n (%)	305 (16.1)	249 (15.9)	56 (17.1)	0.596
EF ≤ 40%, n (%)	670 (35.4)	542 (34.6)	128 (39)	0.126
LA (mm), mean (SD)	n = 1863 47 (7.5)	n = 1541 47.3 (7.5)	n = 322 45.5 (7.4)	<0.001
LA > 40 mm, n (%)	1531 (82.2)	1299 (84.3)	232 (72)	<0.001
LA ≤ 40 mm, n (%)	332 (17.8)	242 (15.7)	90 (28)	
Reason for hospitalisation, n (%)				
Electrical cardioversion	72 (2.9)	68 (3.4)	4 (0.9)	0.006
Planned coronary angiography/PCI or ACS	395 (16.2)	262 (13.1)	133 (30.5)	<0.001
Planned CIED implantation/reimplantation	616 (25.2)	493 (24.6)	123 (28.2)	0.115
Heart failure	729 (29.9)	622 (31)	107 (24.6)	0.007
Ablation	19 (0.8)	17 (0.9)	2 (0.5)	0.555 *
Other	457 (18.7)	408 (20.3)	49 (11.2)	<0.001
AF without any procedures	153 (6.3)	135 (6.7)	18 (4.1)	0.042

Data are presented as number (percentage) or mean (standard deviation) (SD), or median (interquartile range). Abbreviations: ACS, acute coronary syndromes; AF, atrial fibrillation; CABG; coronary artery bypass grafting; CAD, coronary artery disease; CIED, cardiac implantable electronic device; COPD, chronic obstructive pulmonary disease; eGFR, estimated Glomerular Filtration Rate; OAC, oral anticoagulants; PAD, peripheral artery disease; PCI, percutaneous coronary interventions; SD, standard deviation; TIA, transient ischaemic attack. *—Fisher’s exact test was used.

**Table 2 ijerph-20-06145-t002:** The comparison of OAC prescriptions over 2004–2011 and 2012–2019.

	OAC 2004–2011 n = 470	OAC 2012–2019 n = 1535	*p*	Non-OAC 2004–2011 n = 281	Non-OAC 2012–2019 n = 155	*p*
Sex (female), n (%)	270 (57.4)	773 (50.4)	0.007	153 (54.4)	74 (47.7)	0.180
Age, mean (SD)	77.2 (6.1)	79 (7.5)	<0.001	79.3 (6.3)	80.1 (7.9)	0.078
Age < 65, n (%)	16 (3.4)	42 (2.7)	0.450	7 (2.5)	6 (3.9)	0.557 *
Age 65–74, n (%)	103 (21.9)	305 (19.9)	0.335	34 (12.1)	22 (14.2)	0.532
Age > 74, n (%)	351 (74.7)	1188 (77.4)	0.223	240 (85.4)	127 (81.9)	0.341
Clinical characteristics, n (%)						
Heart failure, n (%)	389 (82.8)	1316 (85.7)	0.115	252 (89.7)	141 (91)	0.666
Arterial hypertension, n (%)	433 (92.1)	1428 (93)	0.508	256 (91.1)	142 (91.6)	0.857
Vascular disease, n (%)	247 (52.6)	1170 (76.2)	<0.001	163 (58)	117 (75.5)	<0.001
Diabetes mellitus, n (%)	232 (49.4)	825 (53.7)	0.096	142 (50.5)	89 (57.4)	0.168
Previous stroke, n (%)	181 (38.5)	454 (29.6)	<0.001	96 (34.2)	42 (27.1)	0.129
Previous TIA, n (%)	24 (5.1)	102 (6.6)	0.229	8 (2.8)	10 (6.5)	0.070
Peripheral thromboembolic events, n (%)	34 (7.2)	59 (3.8)	0.002	9 (3.2)	3 (1.9)	0.551 *
Myocardial infarction, n (%)	131 (27.9)	587 (38.2)	<0.001	71 (25.3)	64 (41.3)	0.001
PCI, n (%)	70 (14.9)	394 (25.7)	<0.001	32 (11.4)	34 (21.9)	0.003
CABG, n (%)	33 (7)	183 (11.9)	0.003	6 (2.1)	16 (10.3)	<0.001
PAD, n (%)	14 (3)	273 (17.8)	<0.001	10 (3.6)	24 (15.5)	<0.001
COPD, n (%)	49 (10.4)	153 (10)	0.773	40 (14.2)	16 (10.3)	0.243
Bleeding, n (%)	8 (1.7)	65 (4.2)	0.010	10 (3.6)	13 (8.4)	0.031
Peptic ulcer disease, n (%)	12 (2.6)	50 (3.3)	0.440	12 (4.3)	9 (5.8)	0.473
Cancer, n (%)	18 (3.8)	67 (4.4)	0.614	20 (7.1)	16 (10.3)	0.244
Thrombocythemia, n (%)	72 (15.3)	275 (17.9)	0.193	58 (20.6)	30 (19.4)	0.749
Anaemia, n (%)	77 (16.4)	444 (28.9)	<0.001	63 (22.4)	63 (40.6)	<0.001
Dialysis, n (%)	3 (0.6)	1 (0.1)	0.042 *	2 (0.7)	0 (0)	0.541 *
Type of AF, n (%)						
Paroxysmal, n (%)	163 (34.7)	604 (39.4)	0.068	143 (50.9)	72 (46.4)	0.375
Persistent, n (%)	18 (3.8)	108 (7)	0.012	3 (1.1)	6 (3.9)	0.076 *
Permanent, n (%)	289 (61.5)	823 (53.6)	0.003	135 (48)	77 (49.7)	0.744
Non-permanent (paroxysmal + persistent), n (%)	181 (38.5)	712 (46.4)	0.003	146 (52)	78 (50.3)	0.744
Thromboembolic risk						
CHADS2, mean (SD)	4 (0.8)	3.9 (0.9)	0.001	4 (0.9)	3.9 (0.9)	0.667
CHA_2_DS_2_-VASc, mean (SD)	6.1 (0.9)	6.1 (1)	0.792	6.1 (1)	6.1 (1)	0.371
CHA_2_DS_2_-VASc = 5, n (%)	128 (27.2)	468 (30.5)	0.177	83 (29.5)	44 (28.4)	0.800
CHA_2_DS_2_-VASc = 6, n (%)	224 (47.7)	635 (41.4)	0.016	129 (46)	64 (41.3)	0.353
CHA_2_DS_2_-VASc = 7, n (%)	88 (18.7)	269 (17.5)	0.552	47 (16.7)	32 (20.6)	0.309
CHA_2_DS_2_-VASc = 8, n (%)	26 (5.5)	114 (7.4)	0.158	14 (5)	11 (7.1)	0.363
CHA_2_DS_2_-VASc = 9, n (%)	4 (0.9)	49 (3.2)	0.006	8 (2.8)	4 (2.6)	1.000
Bleeding risk						
HAS-BLED, mean (SD)	2.4 (0.6)	2.5 (0.7)	<0.001	2.2 (0.5)	2.6 (0.9)	<0.001
HAS-BLED ≥ 3, n (%)	173 (36.8)	735 (47.9)	<0.001	78 (27.8)	81 (52.3)	<0.001
HAS-BLED ≥ 5, n (%)	0 (0)	12 (0.8)	0.080 *	0 (0)	1 (0.6)	0.356 *
Laboratory test results						
eGFR (mL/min/1.73m^2^) mean (SD)	n = 469 49.1 (15.6)	n = 1528 49.3 (15.7)	0.931	n = 281 49.5 (17.3)	n = 151 47.2 (16.9)	0.155
eGFR ≥ 60 mL/min/1.73m^2^, n (%)	108 (23)	351 (22.9)	0.980	69 (24.5)	29 (19.2)	0.206
eGFR 59–45 mL/min/1.73m^2^, n (%)	163 (34.8)	556 (36.4)	0.519	106 (37.7)	50 (33.1)	0.342
eGFR 44–30 mL/min/1.73 m^2^, n (%)	146 (31.1)	468 (30.6)	0.837	71 (25.3)	51 (33.8)	0.061
eGFR 29–15 mL/min/1.73 m^2^, n (%)	48 (10.2)	140 (9.2)	0.487	32 (11.4)	17 (11.3)	0.968
eGFR < 15 mL/min/1.73 m^2^, n (%)	4 (0.9)	13 (0.9)	1.000	3 (1.1)	4 (2.6)	0.245 *
Echocardiography						
EF (%), mean (SD)	n = 289 47.1 (13)	n = 1278 45.6 (12.8)	0.067	n = 194 45.1 (11.7)	n = 134 45.4 (11.3)	0.721
EF ≥ 50%, n (%)	156 (54)	620 (48.5)	0.093	83 (42.8)	61 (45.5)	0.623
EF 41–49%, n (%)	39 (13.5)	210 (16.4)	0.217	34 (17.5)	22 (16.4)	0.793
EF ≤ 40%, n (%)	94 (32.5)	448 (35.1)	0.414	77 (39.7)	51 (38.1)	0.766
LA (mm), mean (SD)	n = 285 46.1 (7.6)	n = 1256 47.6 (7.5)	0.003	n = 188 44.6 (7.1)	n = 134 46.7 (7.2)	0.003
LA > 40 mm, n (%)	223 (78.2)	1076 (85.7)	0.002	125 (66.5)	107 (79.9)	0.008
LA ≤ 40 mm, n (%)	62 (21.8)	180 (14.3)		63 (33.5)	27 (20.1)	
Reason for hospitalisation, n (%)						
Electrical cardioversion, n (%)	1 (0.2)	67 (4.4)	<0.001	1 (0.4)	3 (1.9)	0.131 *
Planned coronary angiography/PCI or ACS, n (%)	89 (19)	173 (11.3)	<0.001	97 (34.5)	36 (23.2)	0.014
Planned CIED implantation/reimplantation, n (%)	193 (41.1)	300 (19.5)	<0.001	99 (35.2)	24 (15.5)	0.011
Heart failure, n (%)	105 (22.3)	517 (33.7)	<0.001	58 (20.7)	49 (31.6)	<0.001
Ablation, n (%)	2 (0.4)	15 (1)	0.389 *	0 (0)	2 (1.3)	0.126 *
Other, n (%)	47 (10)	361 (23.5)	<0.001	17 (6)	32 (20.6)	<0.001
AF without any procedures, n (%)	33 (7)	102 (6.6)	0.776	9 (3.2)	9 (5.8)	0.191

Data are presented as number (percentage) or mean (standard deviation) (SD). Abbreviations: ACS, acute coronary syndromes; AF, atrial fibrillation; CABG; coronary artery bypass grafting; CAD, coronary artery disease; CIED, cardiac implantable electronic device; COPD, chronic obstructive pulmonary disease; eGFR, estimated Glomerular Filtration Rate; OAC, oral anticoagulants; PAD, peripheral artery disease; PCI, percutaneous coronary interventions; SD, standard deviation; TIA, transient ischaemic attack. *—Fisher’s exact test.

**Table 3 ijerph-20-06145-t003:** The probability of not using OAC based on sociodemographic and clinical characteristics (multivariable model).

	OR	95% CI	*p*
Age > 74	1.78	1.32–2.42	<0.001
Hospitalisation (in 2004–2011 vs. 2012–2019)	0.17	0.14–0.22	<0.001
Clinical characteristics			
Heart failure	2.05	1.41–2.98	<0.001
Vascular disease	0.97	0.75–1.25	0.805
Cancer	2.16	1.38–3.40	0.001
Type of AF			
Paroxysmal	1.69	1.34–2.14	<0.001
Bleeding risk			
HAS-BLED ≥ 3	1.02	0.80–1.30	0.859
Reason for hospitalisation			
Electrical cardioversion	0.88	0.31–2.52	0.815
Planned coronary angiography/PCI or ACS	2.41	1.81–3.22	<0.001
Planned CIED implantation/reimplantation	1.21	0.85–1.48	0.420

Abbreviations: ACS, acute coronary syndromes; AF, atrial fibrillation; CIED, cardiac implantable electronic device; CI, confidence interval; OAC, oral anticoagulants; OR, odds ratio; PCI, percutaneous coronary interventions.

## Data Availability

Data are available on request for the first author.

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
