# Peer review of "Anticoagulant Treatment in Patients with AF and Very High Thromboembolic Risk in the Era before and after the Introduction of NOAC: Observation at a Polish Reference Centre"

_ijerph, 2023, doi:10.3390/ijerph20126145_

Round 1

Reviewer 1 Report

Dear author;

I reviewed the article entitled ‘Anticoagulant treatment in patients with AF and very high thromboembolic risk in the era before and after the introduction of NOAC: Observation at a Polish reference centre’. I found the article very interesting and useful for our journal. However, I have some minor comments before the acceptance of article.

--minor comments;

1-There are some grammatical mistakes in the article, hence, I recommend the proof-reading

2- The results section in the article is a lit bit long; therefore, it should be decreased.

3- Lastly, please cite this article: Novel electrocardiography parameter for paroxysmal atrial fibrillation in acute ischaemic stroke patients: P wave peak time.Öz A, Cinar T, Kızılto Güler C, Efe SÇ, Emre U, KarabaÄŸ T, Ayça B. Postgrad Med J. 2020 Oct;96(1140):584-588.

Author Response

Proszę zobaczyć załącznik.

Reviewer 2 Report

Thank you for your article. Even though the large database corresponds to a single center, it can provide us with valuable insight into the management of a particular pathology in a large group of patients. However, I have some minor (1-4) and major (5-6) concerns that need to be addressed before it can be published.

1. L45. Non-changeable is not the best way to describe unmodified factors

2. L113. Correct ENDOXABAN for EDOXABAN.

3. Please check the numbers in Table 1. For example, in the 'Type of AF' section, the percentages in the 'Total' column add up to more than 100%. To clarify the information, try expressing the row data as a percentage of the total in the first column, rather than as a percentage of the overall total. The same for Table 2.

4. Can you state why you think NOACs are more commonly prescribed than VKAs for this group of patients today?

5. While it is an extensive database, the results of the associations are not surprising, and the conclusion is obvious. Could you indicate any utility beyond the description of data and associations known since years ago?

6. I am not sure if your clinical question is the best because there are other articles published that involved the same topic, for example: PMID 31005934, PMID: 30928922. What can you add to give an extra to the readers?

Reviewer 3 Report

The submitted manuscript presents data from a retrospective study aiming to analyze therapeutic methods of stroke prevention in very high-risk patients in the period before and after the introduction of NOAC. It’s relevant for the field, scientifically sound, and presented in a well-structured manner. The performed statistical analysis is convincing. I have an insignificant remark to the "new" anticoagulants on the lines 69 and 114. Maybe it's time to stop calling them new. The decoding of the abbreviation NOAC was introduced absolutely correctly (line 39). So, the authors may be right in this context and the correction of this remark is at their discretion. In conclusion, the work has been done clearly and provides relevant results. I have no significant remarks. The manuscript may be recommended for publication.

Reviewer 4 Report

The manuscript is very interesting, methodology is correct, and the main strength is the sample size of included patients.

However some major issues emerged after reading the manuscript.

MAJOR ISSUES

Line 32-33: the conclusion of the abstract is misleading. I suggest to change “can improve our understanding” with “outline reasons concerning…”.

No data are reported as follow-up. Patients discharged without prescription of OAC could be hypothetically evaluated after, e.g., 1-3 months, since the presence of a temporary contraindication to OAC should be re-evaluated in the ambulatory settings. It is difficult to believe that those patients without a prescription of OAC, in the presence of AF, are not evaluated after some times in ambulatory settings for being re-evaluated to initiate OAC treatment.

As an example, think about the following:

A well-known reason for not-prescribing OAC is the occurrence of hemorrhagic stroke, or the presence of acute/subacute ischemic stroke, whose extension is correlated with the risk of hemorrhagic transformation. Since no data are reported regarding the follow-up of those patients (stroke patients discharged without OAC, are evaluated after, e.g., 1 month for introduction of OAC?), it should be outlined in the paper.

Line 78: the cut-off of 5 and 6 points on CHADVASC are used to define “very high risk” patients. Why the authors decided to use these cut-off to define high risk patients?

Line 87: usually thrombocytopenia is defined as platelet count less than 100’000.

Lines 85-96: how were data collected? Manually? Electronically? Please add more details.

Line 111: why some patients are discharged on LMWH? Usually this therapy is a temporary therapy before shifting to AVK, or for those patients with contraindications for OAC. For how many days was LMWH prescribed?

Line 122: what the authors mean by “significant discrepancies”?

Line 126: p-values less than 0.05?

Line 130-132: these are results, not methods.

Table 1: in the main-text define the type of AF reported (paroxysmal, persistent, permanent) and especially the meaning of non-permanent.

Line 206-208: was there also a shift of patients on AVK to NOAC? Or the full sample was composed of different patients?

MINOR ISSUES

Keywords should be listed in alphabetical order.

Please note that some acronyms are not defined (e.g. CABG), or they are defined in tables but not in the main-text. Please double-check the manuscript.

Line 113: correct Endoxaban.

Round 2

Reviewer 4 Report

Authors answered to all my previous comments. I have no further issues.

Author Response

Proszę zobaczyć załącznik.
